# Interaction between Bacteria and the Immune System for Cancer Immunotherapy: The α-GalCer Alliance

**DOI:** 10.3390/ijms23115896

**Published:** 2022-05-24

**Authors:** Arsenij Ustjanzew, Valentin Sencio, François Trottein, Jörg Faber, Roger Sandhoff, Claudia Paret

**Affiliations:** 1Institute of Medical Biostatistics, Epidemiology and Informatics (IMBEI), University Medical Center of the Johannes Gutenberg University Mainz, 55131 Mainz, Germany; arsenij.ustjanzew@uni-mainz.de; 2Centre d’Infection et d’Immunité de Lille, Inserm U1019, CNRS UMR 8204, University of Lille, CHU Lille, Institut Pasteur de Lille, 59000 Lille, France; valentin.sencio@gmail.com (V.S.); francois.trottein@pasteur-lille.fr (F.T.); 3Department of Pediatric Hematology/Oncology, Center for Pediatric and Adolescent Medicine, University Medical Center of the Johannes Gutenberg University Mainz, 55131 Mainz, Germany; faber@uni-mainz.de; 4Lipid Pathobiochemistry, German Cancer Research Center, 69120 Heidelberg, Germany; r.sandhoff@dkfz-heidelberg.de

**Keywords:** microbiome, microbiota, immunotherapy, alpha-galactosylceramide, iNKT, colorectal cancer, biomarkers

## Abstract

Non-conventional T cells, such as γδ T and invariant natural killer T (iNKT) cells, are emerging players in fighting cancer. Alpha-galactosylceramide (α-GalCer) is used as an exogenous ligand to activate iNKT cells. Human cells don’t have a direct pathway producing α-GalCer, which, however, can be produced by bacteria. We searched the literature for bacteria strains that are able to produce α-GalCer and used available sequencing data to analyze their presence in human tumor tissues and their association with survival. The modulatory effect of antibiotics on the concentration of α-GalCer was analyzed in mice. The human gut bacteria *Bacteroides fragilis*, *Bacteroides vulgatus,* and *Prevotella copri* produce α-GalCer structures that are able to activate iNKT cells. In mice, α-GalCer was depleted upon treatment with vancomycin. The three species were detected in colon adenocarcinoma (COAD) and rectum adenocarcinoma tissues, and *Prevotella copri* was also detected in bone tumors and glioblastoma tissues. *Bacteroides vulgatus* in COAD tissues correlated with better survival. In conclusion, α-GalCer-producing bacteria are part of the human gut microbiome and can infiltrate tumor tissues. These results suggest a new mechanism of interaction between bacteria and immune cells: α-GalCer produced by bacteria may activate non-conventional T cells in tumor tissues, where they can exert a direct or indirect anti-tumor activity.

## 1. Introduction

Immunotherapy with conventional T cells is one of the most analyzed and executed approaches in cancer treatment. However, non-conventional T cells, such as γδ T cells and Vα24-invariant natural killer T (iNKT) cells, can also promote tumor rejection. Infiltration of γδ T cells and iNKT cells has been associated with good prognosis, and several clinical studies are analyzing their efficacy in the eradication of tumors (reviewed in [1]). Non-conventional T cells recognize lipids via CD1d, and the glycolipid α-galactosylceramide (α-GalCer) is currently used in several clinical studies as an exogenous ligand to harness iNKT cells against cancer cells. α-GalCer has been isolated from a marine sponge. Although faint amounts of endogenous α-GalCer have been found in human immune cells [2,3], mammalian enzymes catalyzing the production of α-anomeric GalCer are unknown. A synthetic version of α-GalCer (KRN7000) is used as a potent activator of iNKT cells and as a model CD1d antigen. iNKT cells recognize α-GalCer displayed by CD1d and are specifically activated to rapidly produce interferon-γ (IFN-γ) and interleukins, regulating NK cells, CD8+ T cells, and dendritic cells (DCs). γδ T cells do not recognize the CD1d/α-GalCer complex directly, but α-GalCer administration leads to IFN-γ production by γδ T cells, probably through a bystander effect involving iNKT cells [4]. iNKT cells have the capacity to mediate direct cytolytic activity against CD1d positive tumor cells via perforin and granzyme B. Even if the majority of solid tumors are CD1d-negative, iNKT cells can suppress tumor growth indirectly by targeting CD1d-positive elements of tumor-supportive stroma, such as tumor-associated macrophages [5]. Moreover, the epigenetic induction of CD1d in tumors has been suggested as a mechanism to sensitize lung cancer to iNKT treatment [6], and thymosin alpha1, which is widely used as an immune adjuvant to cancer therapy, was observed to enhance the cytotoxicity of iNKT cells against colon cancer via upregulating CD1d expression [7].

Modulation of the immune system for the eradication of cancer is a complex process involving different players of the tumor microenvironment. Intestinal and tumor resident bacteria are important regulators of this process. The abundance of specific microbes correlates with the density of anticancer T cells or suppressive immune cells [8,9]. Several mechanisms have been discussed, such as the presentation of bacterial peptides by cancer cells and immune cells, bacterial antigen mimicry with tumor antigens, and the effect of microbe-derived metabolites (for a review, see [10]). Importantly, modulation of the microbiome via antibiotics may affect the efficacy of immunotherapy [11].

The primary habitat of the human commensal microbiota is the gut, but microbial populations also exist in skin, oral, respiratory, and genital tracts [12]. While interaction between bacteria and the immune system is generally analyzed in the context of the intestinal microbiome, bacteria can also infiltrate tissues of different tumor entities [13,14]. The intratumor microbiome has been associated with the infiltration of cytotoxic CD8+ T cells and patient survival in cutaneous melanoma [9]. In this context, intratumor-residing gut microbiota have been suggested to modulate chemokine levels and affect CD8+ T cell infiltration, consequently influencing patient survival. In pancreatic cancer, and in contrast to normal pancreatic tissue, bacteria and fungi colonize the cancerous tissues, affecting the response and prognosis to treatment [15]. Interestingly, bacterial infection within or near the tumor bed has also been suggested to stimulate patients’ immune responses in brain tumors, for example, in glioblastoma (GBM) patients [16].

Here, we question if α-GalCer, the potent activator of iNKT cells, can function as an endogenous ligand in the gut and in tumor tissues and propose a so-far poorly characterized interaction between bacteria and the immune system that may have particular relevance in the context of immunotherapy.

## 2. Results

### 2.1. α-GalCer Is Produced by Human and Mouse Commensal Bacteria

α-GalCer was isolated from a marine sponge, and a synthetic version of α-GalCer (KRN7000), synthesized for the first time in 1995 [17], is used as a potent activator of iNKT cells (Figure 1A). Cells of the human body produce β-GalCer in the glycosphingolipid (GSL) metabolic pathway (Figure 1B,C) instead of α-GalCer (for a review, see [18]). Laura C. Wieland Brown and colleagues have previously shown that α-GalCer can be produced by *Bacteroides fragilis* (α-GalCerBf) [19]. This finding was later confirmed by our group and extended to two other strains of the human gut microbiome, *Bacteroides vulgatus* and *Prevotella copri* [20]. Compared to KRN7000, α-GalCerBf has a shorter N-acyl chain, lacks a hydroxyl group at C4 of the sphingoid base, and has an iso-branched lipid terminus (Figure 1D). α-GalCerBf has been shown to expand iNKT cells in vitro [19] and to stimulate IFN-γ and IL-2 production by iNKT cells in vitro and in vivo [19,21]. α-GalCer was also previously detected by our group in the large intestine of mice, but it is not clear which bacterial strains are required for its production [21]. The identified α-GalCer contained a β(R)-hydroxylated hexadecanoyl chain N-linked to C18-sphinganine, which differed from what has been reported with *Bacteroides fragilis* (Figure 1E). This α-GalCer was also able to induce iNKT cell activation in vitro [21]. Here, we further determined whether the concentration of α-GalCer in mouse guts is influenced by the presence of antibiotics. Mice were treated with vancomycin or with colistin alone or in combination. After 3 weeks of treatment, the presence of α-GalCer or host β-GalCer and β-GlcCer in the feces, caecum, and liver of mice was analyzed. As shown in Figure 2, only α-GalCer but no β-anomeric HexCer (β-GlcCer or β-GalCer) is depleted from the host after treatment with vancomycin. Colistin has no effect on α-GalCer content.

In summary, different forms of α-GalCer produced by commensal bacteria of the human and mouse gut have been described in the literature. Previous works indicate that these α-GalCer structures are able to induce iNKT cell activation. In addition, we show here that the presence of α-GalCer is modulated by antibiotics.

### 2.2. α-GalCer-Producing Bacteria Infiltrate Human Tumors

We reanalyzed the data published in the article of [22] and in The Cancer Microbioma Atlas (TCMA) [13] to determine whether *Bacteroides vulgatus*, *Bacteroides fragilis,* and *Prevotella copri* can infiltrate human tumors. In [22], the authors undertook a comprehensive analysis of the tumor microbiome, studying 1526 tumors and their adjacent normal tissues across seven cancer types, including colon, breast, lung, ovary, pancreas, melanoma, bone, and brain tumors. TCMA is a public collection of microbial compositions of oropharyngeal, esophageal, gastrointestinal, and colorectal tissues. As summarized in Table 1, *Bacteroides vulgatus*, *Bacteroides fragilis*, and *Prevotella copri* were found in 40%, 45%, and 27% of colorectal carcinoma tissue samples, respectively. As expected, all three were also present in the normal colon, but *Bacteroides fragilis* and *Prevotella copri* were found more frequently in tumor samples than in normal adjacent tissues. *Prevotella copri* was also detected in 10% of bone tumors and GBM and in circa 9% of breast and ovary tumors. Lower infiltration of the three species in other tumor entities was observed but has to be carefully evaluated due to a possible contamination of the samples, which was particularly evident for *Prevotella copri* (circa 4% of extraction negative controls were positive).

Infiltration of the three species was also confirmed in the whole-exome sequencing (WXS) data of TCMA (Figure 3). Of the five tumor entities analyzed in the atlas, infiltration was found in colon adenocarcinoma (COAD) and rectum adenocarcinoma (READ), but not in head and neck squamous cell carcinoma, esophageal carcinoma, or stomach adenocarcinoma. As shown in Table 2, *Bacteroides vulgatus* and *Bacteroides fragilis* were more abundant than *Prevotella copri*. COAD was more infiltrated than READ.

In summary, α-GalCer-producing bacteria infiltrate several tumor tissues, not only in the digestive tract but also, to a lesser extent, tumors in bone and the brain.

### 2.3. The Infiltration of Bacteroides vulgatus Is Associated with the Overall Survival (OS) in COAD

The microbiome may have prognostic significance. We used the survival data included in TCMA to analyze a possible correlation between the infiltration of the three bacteria in tumor tissues and the OS in COAD and READ. We found a possible significant (*p* = 0.037) OS advantage in patients with COAD tumor stage IV and the presence of *Bacteroides vulgatus* (Figure 4). Note that the confidence limits of the curves are relatively wide (not shown), making meaningful interpretations difficult. Appendix A shows a complete table of comparisons and *p*-values.

### 2.4. Suggested Model of the Interaction between Bacteria and Tumors

Based on our findings, we suggest a new mechanism for how bacteria can modulate the microenvironment and contribute to the successful eradication of a tumor (Figure 5). α-GalCer-producing bacteria infiltrate tumor tissues where they are engulfed by antigen presenting cells (APCs). Lipids of microbial origin are processed and loaded on CD1d inside APCs and are thereafter presented on the cell surface [23]. APCs can therefore present α-GalCer to iNKT cells. Activated iNKT cells produce large amounts of pro-inflammatory (IFN-γ) and immune regulatory (IL-4) cytokines. α-GalCer also induces IFN-γ production by γδ T cells via an indirect mechanism involving iNKT cells. These events induce the activation of CD8+ T cells, thus enhancing other immune cell responses to indirectly eradicate tumors. Moreover, bacteria can enter tumor cells, and antigens derived from intracellular bacteria can be presented by tumor cells and activate the immune system [24]. Therefore, CD1d-expressing tumors may directly present α-GalCer to iNKT cells. iNKT cells produce granzymes (Gzms) A and B and can recognize and lyse CD1d-expressing tumors. These processes are regulated by antibiotics modulating the microbiome composition and, therefore, the amount of α-GalCer.

## 3. Discussion

Over a hundred years ago, William B Coley, the “Father of Immunotherapy”, showed that the injection of bacteria into tumors can induce their shrinkage by stimulating the immune system to attack the tumors. We now know that intestinal and tumor resident bacteria are an important modulator of the immune system, but not all mechanisms have yet been clarified.

The reason why unconventional T cells infiltrate and eradicate tumors is not fully understood. Several endogenous lipid ligands have been suggested to be presented via CD1d to unconventional T cells. These include phospholipids and, so far, unknown tumor-specific lipids [25]. However, α-GalCer represents the most potent activator of iNKT cells, and, so far, only exogenous α-GalCer has been used for stimulation in preclinical and clinical studies. Known human glycosylceramide synthases (encoded by the genes *UGCG* and *UGT8*) can only produce β-GlcCer and β-GalCer, but previous studies disclose that, besides sponges, the bacteria of the strains *Bacteroides fragilis*, *Bacteroides vulgatus,* and *Prevotella copri* can also produce α-GalCer. *Prevotella* and *Bacteroides* are thought to have a common ancestor and are present in the human gut. Either *Prevotella* or *Bacteroides* dominates the gut, and both have been theorized to be antagonistic. *Prevotella* is more common in populations consuming a plant-rich diet. These bacteria may represent a source of α-GalCer in the gut or in tumor tissues and may play a role in the infiltration and activation of non-conventional T cells, particularly in tumors of the digestive tract.

In colorectal cancer (CRC), the abundance of *Bacteroides fragilis* has been shown to increase in tumors compared to healthy tissue [26,27], which is in accordance with our results. Our data suggest the relevance of two further species, *Bacteroides vulgatus* and *Prevotella copri*, in the biology of CRC. Importantly, our data show a positive correlation between the infiltration of *Bacteroides vulgatus* and better survival in stage IV COAD. Stage IV indicates that cancer has spread to other tissues and organs and that outcomes for these patients are very poor [28]. There are few established prognostic factors for stage IV COAD, including the mutation status of *BRAF* and *KRAS* [29,30,31], and histological subtype [32]. Stratification factors for future studies evaluating new cancer treatments, including immunotherapy, are required. Immune checkpoint inhibitors (ICIs) have exhibited a clinical benefit only in CRC patients with deficient mismatch repair (dMMR)/high levels of microsatellite instability (MSI-H), comprising approximately only 5% of metastatic CRC (mCRC) cases, and some do not respond to ICI treatment. Moreover, the remaining 95% of CRCs often exhibit a lower tumor mutation burden, limiting the efficacy of immunotherapies based on conventional T cells that recognize mutated epitopes. Current efforts address the targeting of common genetic alterations such as *TP53* mutations, which are found in 70% of COAD [33], by humoral and cell-based immunotherapies, including T cell receptor mimic monoclonal antibodies [34]. Our data suggest that patients with an infiltration of α-GalCer-producing bacteria in the tumor tissues could benefit from immunotherapy based on non-conventional T cells. Increased infiltration of iNKT cells has been described in CRC. Patients with high iNKT cell infiltration showed overall higher as well as disease-free survival rates. Va24+ NKT cell tumor infiltration is considered an independent positive prognostic factor in human CRC [35]. Interestingly, Lee and collaborators found that *Bacteroides fragilis* shows a protective effect in a mouse model for colitis-associated CRC and suggested the use of *Bacteroides fragilis* as a preventive therapeutic strategy against inflammatory CRC [36]. If the production of α-GalCer by infiltrating bacteria can activate iNKT cells to control the tumor and therefore improve the patients’ survival, it is worth analyzing it in future studies. 

Our model suggests a new mechanism of interaction between commensal bacteria and the immune system. α-GalCer produced by the bacteria can be presented via APCs or tumor cells to iNKT cells, leading to a direct or indirect eradication of tumors. Data from the literature suggest a correlation between the presence of the three bacteria in the gut and the answer to immunotherapy. The recolonization of *Bacteroides fragilis* in antibiotic-treated mice was previously observed to rescue CTLA-4 blockade resistance, and oral gavage of *Bacteroides fragilis* induced the Th1 immune response and DC maturation in the tumor-draining lymph node [37]. In NSCLC, *Prevotella copri* was observed to be enriched in the gut of patients responding to treatment with nivolumab [38]. Huang et al. demonstrated that *Bacteroides vulgatus* of the gut microbiome is associated with an improved response to anti-PD-1 blockade treatment in a mouse model and in NSCLC patients, and they suggested that *Bacteroides vulgatus* can be used as an adjuvant to anti-PD-1 immunotherapy [39]. Importantly, our data show that the presence of α-GalCer is modulated by antibiotics. Several studies have shown that antibiotics negatively impact the effectiveness of immune checkpoint inhibitors, and the perturbation of the gut microbiota has been indicated as a putative mechanism [11]. The impact of the early use of vancomycin exposure and outcomes from ICI therapy have been shown, for example, in non-small-cell lung cancer (NSCLC) [40]. Thus, antibiotics may modulate the efficacy of ICI and potentially other immunotherapy approaches by affecting the concentration of α-GalCer.

Several questions remain to be answered. First, the concentration of α-GalCer produced by bacteria in vivo in the gut or in tumor tissues has yet to be analyzed. *Bacteroides fragilis*, *Bacteroides vulgatus*, and *Prevotella copri* do not produce the same quantity of α-GalCer [20]. Moreover, the production of α-GalCer decreases in mice exposed to conditions that alter the composition of the gut microbiota, including the Western diet, colitis, and influenza [21]. Second, the presentation of endogenous α-GalCer produced by bacteria to tissue-resident iNKT cells, even if suggested by our results, still remains to be demonstrated. Future experiments should verify in mouse models how the presence of α-GalCer-producing bacteria in the gut and/or in tumor tissues influences tumor growth in the presence and absence of vancomycin.

In conclusion, the iNKT cell ligand and activator α-GalCer may be present in the gut microbiome or in tumor tissues depending on the bacteria composition. Bacteria-derived α-GalCer in the tumor microenvironment can orchestrate the ability of conventional and non-conventional T cells to infiltrate and eradicate tumors. Further studies are needed to define which factors may influence and modulate the production of α-GalCer in human tumors to increase the effectiveness of immunotherapy or the natural defense of the human body against cancer.

## 4. Materials and Methods

### 4.1. Identification of α-GalCer-Producing Bacteria in Human Tissues

We searched in PubMed for articles describing α-GalCer structures in bacteria. We reanalyzed the data published in the article of [22] and in The Cancer Microbioma Atlas [13] to assess the infiltration of relevant bacteria strains in tumor tissues. In [22], the authors undertook a comprehensive analysis of the tumor microbiome, studying 1526 tumors and their adjacent normal tissues using next-generation sequencing. In Appendix A, a list of all identified bacteria and the clinical information of the patients was made available. TCMA is a public database of decontaminated, tissue-resident microbial profiles of TCGA gastrointestinal cancer tissues. We reanalyzed the data published in [22] by calculating the percentage of samples containing more than zero reads per tissue type for each bacterium of interest. The same calculation was performed with the WXS TCMA data [13] of the relative abundance per project (“COAD”, “READ”, “HNSC”, “ESCA”, and “STAD”) for each bacterium of interest.

### 4.2. Treatment of Mice with Antibiotics

Specific pathogen-free C57BL/6J mice (8-week-old, male) were purchased from Janvier (Le Genest-St-Isle, France). Mice were maintained in the facility in the Animal Resource Centre at the Institut Pasteur de Lille for at least two weeks prior to usage to allow appropriate acclimation. Mice were treated for 3 weeks with vancomycin (Van) (1 g/L), colistin (Col) (1 g/L) or both antibiotics or were not treated (NT) in drinking water. The cages were changed every two days. Feces were collected before starting treatment (T0) and after 3 weeks of treatment (T3). Caecum, content of caecum, and liver were also collected.

### 4.3. Lipid Extraction

Feces, caecum, caecum content, and liver from antibiotic-treated mice and from control mice were extracted for lipids, as described previously [20]. In brief, tissues or feces were homogenized in methanol on ice using the TissueLyser II of Qiagen (Hilden, Germany) and a stainless steel bead (5 mm) per tube. Dried homogenates were extracted 3 times with mixtures of chloroform/methanol/water (2 times with 10/10/1 and then with 30/60/8). Supernatants were pooled, dried, and subjected to methanolic mild alkaline hydrolysis (0.1 M potassium hydroxide) for 2 h at 37 °C, and, subsequently, they were neutralized with acetic acid. Saponified extracts were then desalted using reverse-phase C-18 columns.

### 4.4. Reversed-Phase LC-MS2 Analysis of Glycosylceramides (HexCer) Including Bacterial α-GalCer

Glycosylceramides were analyzed using LC-MS2 according to [20]. In brief, aliquots of lipid extracts corresponding to 10 mg of mammalian tissue dry weight were mixed with internal lipid standards for analysis with LC-MS2 using an Acquity I-class UPLC and a Xevo TQ-S “triple quadrupole” instrument, both from Waters (Eschborn, Germany). Using a CSH C18 column (2.1 × 100 mm, 1.7 µm; Waters), lipids were measured in reversed-phase LC mode with a gradient between 57% solvent A (50% methanol) and 99% solvent B (1% methanol and 99% isopropanol), both containing 0.1% formic acid and 10 mM of ammonium formate as additives. Lipids analyzed were detected using multiple reaction monitoring (MRM). For corresponding transitions, see the supplemental tables in [20]. GlcCer(d18:1;14:0), GlcCer(d18:1;19:0), GlcCer(d18:1;25:0), and GalCer(d18:1;31:0) were used as internal standards to quantify α- and β-anomers of GlcCer and GalCer. Student’s *t*-test was conducted using GraphPad Prism (GraphPad Software, San Diego, CA, USA), and *p* < 0.05 was considered to be statistically significant. Data are presented as the mean ± standard deviation.

### 4.5. Survival Analysis of the TCMA Data

We performed a Kaplan–Meier survival estimation analysis on the WXS TCMA data [13] to examine whether the presence of the species *Bacteroides fragilis*, *Bacteroides vulgatus,* and *Prevotella copri* alone and their presence in combination with the tumor stage were predictive of OS. We compared the survival curves between the conditions “rel. abundance of < bacterium X > is absent” vs. “rel. abundance of < bacterium X > is present”, where < bacterium X > is one of the three bacteria of interest, “absent” equals rel. abundance = 0, and “present” equals rel. abundance ≠ 0, with the log-rank test conducted to assess its individual prognostic value. For this analysis, R packages “survival” [41] and “survminer” [42] were used.

## Figures and Tables

**Figure 1 ijms-23-05896-f001:**
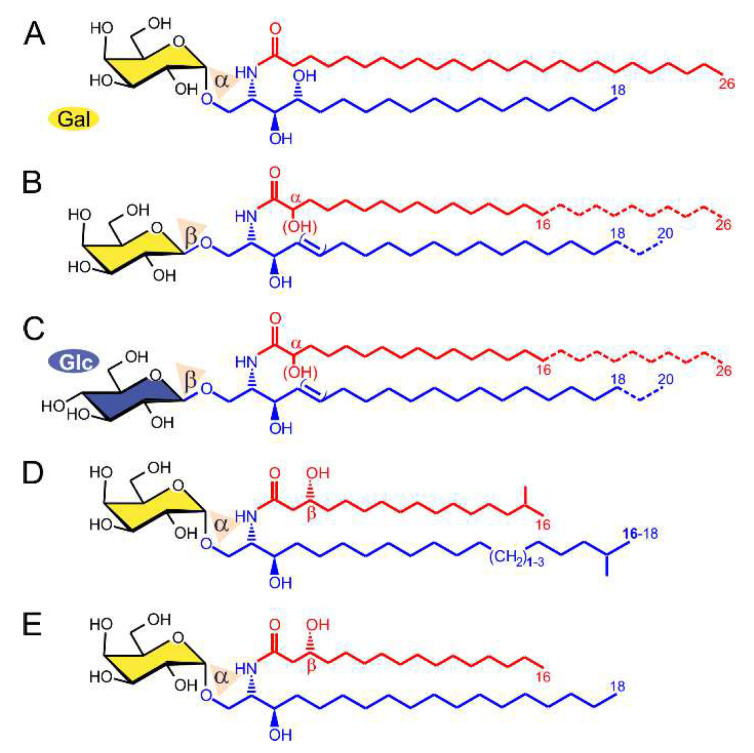
Structures of immunogenic α-GalCer and mammalian β-hexosylceramides. The α-GalCer KRN7000 is a synthetic glycolipid similar to galactosylceramides isolated from the marine sponge *Agelas mauritianus* (**A**). α-GalCer is a strong immunostimulant and shows potent anti-tumor activity in many in vivo models. Humans can produce only the beta-anomeric form of GalCer (**B**), as well as that of glucosylceramide (GlcCer) (**C**), both of which cannot stimulate iNKT cells. Commensal human bacteria can produce a α-GalCer with a shorter ceramide anchor compared to KRN7000. The ceramide anchor consists of an iso-branched sphinganine and an iso-branched beta-hydroxy acyl chain (**D**). Bacteria of the mouse gut can also produce α-GalCer similar to that in humans, but without iso-branching of the sphinganine or acyl chain (**E**). Chain length variations are drawn with dotted lines, while functional groups, which are not necessarily present, are set in brackets.

**Figure 2 ijms-23-05896-f002:**
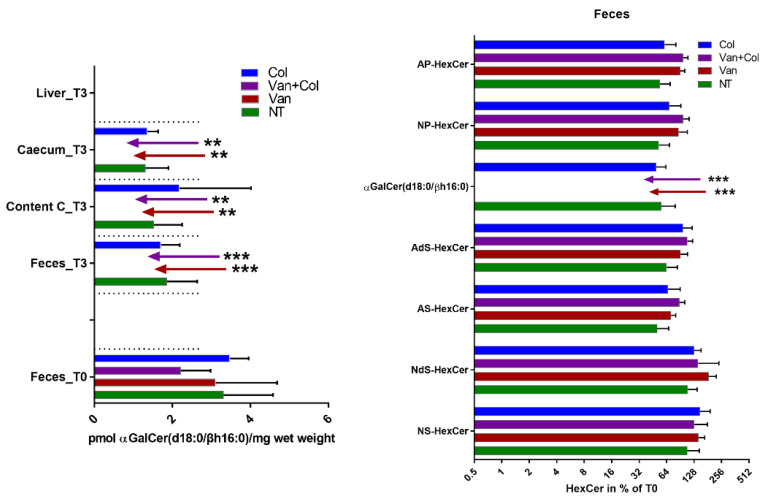
Treatment of mice with the antibiotic vancomycin (Van) led to loss of α-GalCer in feces and intestine of mice. Mice were treated for 3 weeks with vancomycin (Van), colistin (Col), with both antibiotics (1 g/L), or were not treated (NT). Feces were collected before starting treatment (T0), and after 3 weeks of treatment (T3), together with caecum, content of caecum (Content C) and content of liver were analyzed for the presence of α-GalCer and host β-GalCer and β-GlcCer (summarized as HexCer) with various ceramide anchors (A: alpha-hydroxy acyl chain, N: non-hydroxy acyl chain, P: phytosphingosine, S: sphingosine, dS: dihydrosphingosine). Samples from 4 to 5 mice were analyzed per group, and the mean ± standard deviation is plotted. **: *p* < 0.01, ***: *p* < 0.001 compared to corresponding controls (NT).

**Figure 3 ijms-23-05896-f003:**
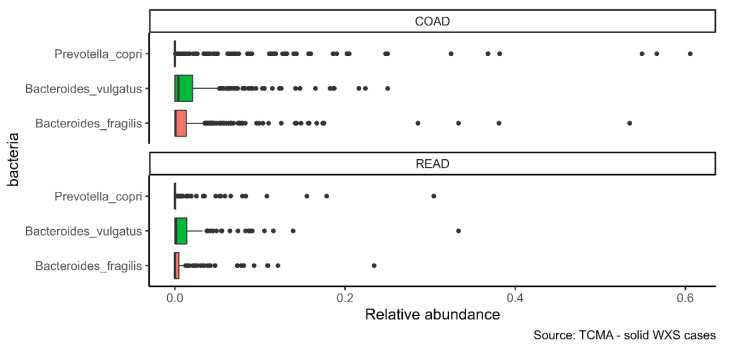
α-GalCer-producing bacteria infiltrate colon and rectum adenocarcinoma tissues. The distribution of the relative abundance of *Bacteroides fragilis*, *Bacteroides vulgatus*, and *Prevotella copri* in colon adenocarcinoma (COAD) and rectum adenocarcinoma (READ) tissues was calculated from TCMA.

**Figure 4 ijms-23-05896-f004:**
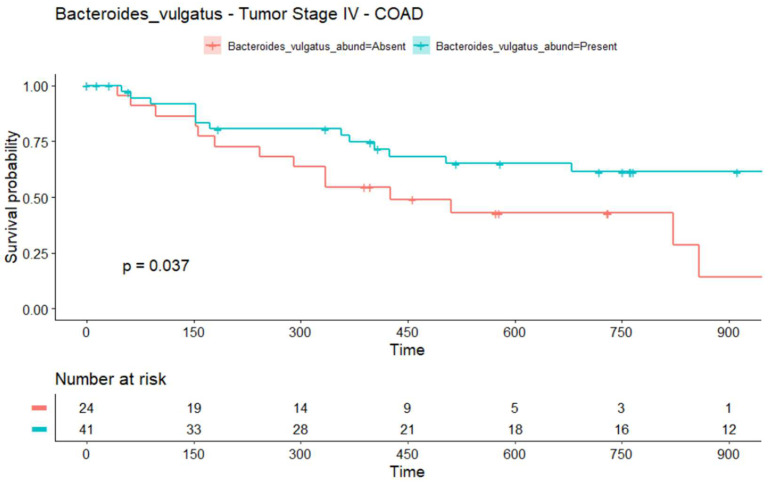
*Bacteroides vulgatus* in COAD tissues is associated with a better survival. OS curves and risk table for *Bacteroides vulgatus* in two conditions—absent (red) and present (blue)—for stage IV colon adenocarcinoma patients. Time is in days.

**Figure 5 ijms-23-05896-f005:**
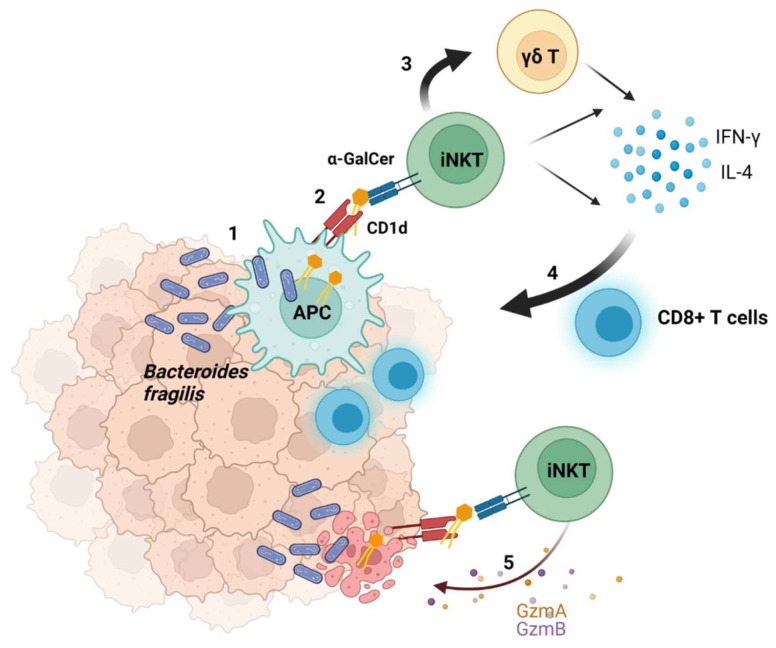
Modulation of the anti-tumor immune response by α-GalCer-producing bacteria. Bacteria infiltrate tumor tissues and are engulfed by antigen presenting cells (APCs) (1). α-GalCer is presented by APCs via CD1d (2) and stimulates iNKT cells to produce lIFN-γ and IL-4. α-GalCer-activated iNKT cells enhance γδ T cell activation and cytokine production (3). These events induce the activation of CD8+ T cells (4). iNKT cells also produce granzymes (Gzms) A and B (5) and may have a direct tumor-killing effect on CD1d-expressing tumors. Illustration was created with BioRender.com (accessed on 22 April 2022).

**Table 1 ijms-23-05896-t001:** Infiltration of *Bacteroides fragilis*, *Bacteroides vulgatus*, and *Prevotella copri* in tumor tissues. Tumor (T), normal adjacent tissues (NAT), normal fibroadenoma (N-FA), and negative controls from [22] were included in the analysis. The % of the positive samples, defined as samples that have more than zero normalized reads of the bacterium, is reported. Contaminants included negative controls that were processed with the samples, including 432 DNA extraction controls and 204 polymerase chain reaction (PCR) no-template controls (NTCs), and 165 paraffin-only samples taken from the margins of the paraffin blocks (without tissue).

Tissue	Samples	*Bacteroides* *vulgatus* (%)	*Bacteroides* *fragilis* (%)	*Prevotella* *copri* (%)
Bone (T)	39	0	0	10.26
Breast (N-FA)	28	0	0	0
Breast (N)	51	1.96	3.92	3.92
Breast (NAT)	168	1.19	0.6	5.36
Breast (T)	338	1.48	1.48	9.47
Colon (NAT)	22	45.45	31.82	13.64
Colon (T)	22	40.91	45.45	27.27
Extraction control	432	0.23	1.62	3.94
GBM (T)	40	0	0	10
Lung (NAT)	231	0.87	1.73	3.03
Lung (T)	243	0.41	2.88	6.58
Melanoma (T)	197	3.55	4.57	2.54
NTC	204	0	1.96	2.45
Ovary (N- fallop)	17	5.88	17.65	23.53
Ovary (NAT)	12	8.33	0	0
Ovary (T)	56	1.79	1.79	8.93
Pancreas (T)	67	5.97	5.97	5.97
Paraffin control	165	0.61	0.61	3.64

**Table 2 ijms-23-05896-t002:** Infiltration of α-GalCer-producing bacteria in colorectal adenocarcinoma. The TCMA relative abundance data were used to calculate the presence or absence of the bacteria in colon adenocarcinoma (COAD) and rectum adenocarcinoma (READ) tissues. A bacterium is classified as present if the sample has a rel. abundance value > 0 and as absent if the rel. abundance value = 0.

Project	Samples	*Bacteroides vulgatus* (%)	*Bacteroides fragilis* (%)	*Prevotella copri* (%)
COAD	439	65.83	55.35	28.47
READ	159	54.09	37.74	27.67

## Data Availability

Not applicable.

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
