# Peer review of "Interaction between Bacteria and the Immune System for Cancer Immunotherapy: The α-GalCer Alliance"

_ijms, 2022, doi:10.3390/ijms23115896_

Round 1

Reviewer 1 Report

Dear Authors, 

First of all I would like to congratulate you for this interesting work. I consider that it provides new avenues on the understanding of the role of microbiome on antitumor immune response. I have only suggested minor commentaries to original manuscript. In spite of further research is required to confirm your model hypothesis about specific tumor associated bacteria - lipid presentation -innate cell cycle immunity activation, this is a novel "pathway" to explore, that could be helpful to develop new strategies mainly in "cold" tumours, like in pMMR/MSS COAD. I encourage you to perform dynamic analysis with tissue / faecal samples from patients.  

Author Response

We thank the reviewer for the appreciation of our work. (Note, that the line numbers correspond to the track-changes mode of word with “Markup: all”.) 

1)     Page 2. Line 61. Please, provide a reference for this statement.

We have added two references (lines 61-62).

2)     Page 6. Line 192. "bei", is this correct?

We carefully read through the manuscript and corrected this typo as well as several others.

3)     Page 12. Line 232. Do you have any information about the potential influence on survival results of other prognostic factors in stage IV COAD, like BRAFmut, sidedness...?  

We searched the literature for other prognostic factors and added this at line 242 in the discussion.

Reviewer 2 Report

The manuscript entitled “Interaction between bacteria and the immune system for cancer immunotherapy: the α-GalCer alliance by Arsenij Ustjanzew et al. explored the association between α-GalCer produced by bacteria in the gut and the activation of iNKT cells in the published datasets. iNKT cells are a type of CD1d-restricted specialized innate T cell population that recognize lipid antigens presented by a transmembrane glycoprotein CD1d and have the capacity to induce strong anti-tumor responses. Some of my questions are below:

  1. In figure 1, have the authors presented their own data or reviewed it from the literature?
  2. Did the authors perform any in-house experiment showing the role of gut microbiota on tumor growth (any tumor discussed in table 1) and the effect of vancomycin on these tumors. It would be interesting to see how iNKT cells respond to these tumors in the presence and absence of α-GalCer producing bacteria.

Author Response

We thank the reviewer for addressing these points for a better understanding of the α-GalCer structures and discussion. (Note, that the line numbers correspond to the track-changes mode of word with “Markup: all”.)  

1) In figure 1, have the authors presented their own data or reviewed it from the literature?

Figure 1 shows the structures of known α-GalCer and ?-Hexosylceramides. Our group has described in previous publications the structures in D and E. In the revised manuscript we have clarified, when the structures were discovered and by whom by adding more references in the results part (lines 86, 89, and 91).

2) Did the authors perform any in-house experiment showing the role of gut microbiota on tumor growth (any tumor discussed in table 1) and the effect of vancomycin on these tumors. It would be interesting to see how iNKT cells respond to these tumors in the presence and absence of α-GalCer producing bacteria.

Concerning in-house experiments, we have not done these experiments so far, but we agree that this would be the next step and have commented this point in the discussion at lines 290 - 292.

Reviewer 3 Report

Arsenij Ustjanzew and colleagues present a quality and well-written expetimental manuscript describing interaction between bacteria and the immune system for cancer immunotherapy: the α-GalCer alliance.

In their study authors questioned if α-GalCer, the potent activator of iNKT cells, can function as an endogenous ligand in the gut and in tumor tissues and propose a so far poorly characterized interaction between bacteria and the immune system which may have a particular relevance in the context of immunotherapy.

Authors searched the literature for bacteria strains able to produce α-GalCer, and used available sequencing data to analyze their presence in human tumor tissues and their association to survival. They analyzed the modulatory effect of antibiotics on α-GalCer concentration in mice. The found that human gut bacteria Bacteroides fragilis, Bacteroides vulgatus and Prevotella copri produce α-GalCer structures able to activate iNKT cells. In mice, α-GalCer was depleted upon treatment with vancomycin. The three species were detected in colon adenocarcinoma (COAD) and rectum adenocarcinoma tissues, and Prevotella copri also in bone tumors and Glioblastoma tissues. Bacteroides vulgatus in COAD tissues correlated with a better survival.

Authors established that α-GalCer is produced by human and mice commensal bacteria, α-GalCer producing bacteria infiltrate human tumors, the infiltration of Bacteroides vulgatus is associated with the overall survival (OS) in COAD.

Finally, authors conclude that he iNKT cells ligand and activator α-GalCer may be present in the gut microbiome or in tumor tissues depending on the bacteria composition. Bacteria-derived α-GalCer in the tumor microenvironment can orchestrate the ability of conventional and non-conventional T cells to infiltrate and eradicate tumors. Further studies are needed to define which factors may influence and modulate the production of α-GalCer in human tumors to increase the effectiveness of immunotherapy or the natural defense of the human body against cancer.

==============================

Other comments:

1) Please check for typos throughout the manuscript.

2) Authors are kindly encouraged to cite the following article that describes certain aspects of targeting tumors using immunotherapies (both innate and adaptive). DOI: 10.3389/fimmu.2021.707734.

Overall, the manuscript is highly valuable for the scientific community and should be accepted for publication after the corrections are made.

Author Response

We thank the reviewer for the very positive assessment of our manuscript! (Note, that the line numbers correspond to the track-changes mode of word with “Markup: all”.)

1) Please check for typos throughout the manuscript.

We have revised the manuscripts and corrected typos.

2) Authors are kindly encouraged to cite the following article that describes certain aspects of targeting tumors using immunotherapies (both innate and adaptive). DOI: 10.3389/fimmu.2021.707734.

We have pointed out the importance of tp53 at line 251 and added the citation at line 253. 

Round 2

Reviewer 2 Report

Accept